# The Crosstalk between FcεRI and Sphingosine Signaling in Allergic Inflammation

**DOI:** 10.3390/ijms232213892

**Published:** 2022-11-11

**Authors:** Hyein Jo, Kyeonghee Shim, Dooil Jeoung

**Affiliations:** Department of Biochemistry, College of Natural Sciences, Kangwon National University, Chuncheon 24341, Korea

**Keywords:** allergic inflammation, FcεRI signaling, micro RNAs, sphingolipids, sphingosine kinase, sphingosine signaling

## Abstract

Sphingolipid molecules have recently attracted attention as signaling molecules in allergic inflammation diseases. Sphingosine-1-phosphate (S1P) is synthesized by two isoforms of sphingosine kinases (SPHK 1 and SPHK2) and is known to be involved in various cellular processes. S1P levels reportedly increase in allergic inflammatory diseases, such as asthma and anaphylaxis. FcεRI signaling is necessary for allergic inflammation as it can activate the SPHKs and increase the S1P level; once S1P is secreted, it can bind to the S1P receptors (S1PRs). The role of S1P signaling in various allergic diseases is discussed. Increased levels of S1P are positively associated with asthma and anaphylaxis. S1P can either induce or suppress allergic skin diseases in a context-dependent manner. The crosstalk between FcεRI and S1P/SPHK/S1PRs is discussed. The roles of the microRNAs that regulate the expression of the components of S1P signaling in allergic inflammatory diseases are also discussed. Various reports suggest the role of S1P in FcεRI-mediated mast cell (MC) activation. Thus, S1P/SPHK/S1PRs signaling can be the target for developing anti-allergy drugs.

## 1. Possible Role of Sphingolipids in Allergic Inflammation

Sphingolipids are present in the cell membrane and provide structural integrity; they can mediate cell signaling, cell survival and differentiation, immune cell trafficking, and vascular and mucosal integrities [1,2]. Sphingomyelin is converted to ceramide by sphingomyelinase (SMase), and ceramide is converted to sphingosine by ceramidase. Sphingosine can be phosphorylated by two forms of sphingosine kinases (SPHK1 and SPHK2) to form S1P [3,4,5,6,7,8,9,10]. S1P can also be degraded by S1P lyase and dephosphorylated by S1P phosphatase. Figure 1 shows the metabolic pathways of sphingolipids. SPHK1 and SPHK2 can be activated by growth factors, cytokines, and other extracellular stimuli to generate S1P [11].

SPHK1 is mainly localized in the cytosol and can be activated by epidermal growth factor (EGF), platelet-derived growth factor (PDGF), vascular endothelial growth factor, hepatocyte growth factor (HGF), and tumor necrosis factor-α (TNF-α) [12]. SPHK2 is localized in the nucleus and mitochondria-associated outer membrane [13,14] and is activated by transforming growth factor-β (TGF-β) [15].

Ceramide-1-phosphate (C1P) promotes caspase-3-mediated apoptosis and cell cycle arrest while S1P is necessary for cell proliferation and survival [16,17]. S1P has garnered attention for its pro-inflammatory effects [18,19] and is a potent bioactive lipid molecule that regulates cell growth, apoptosis, and the immune system [20,21,22,23]. Elevated levels of S1P have been observed in the bronchoalveolar lavage (BAL) fluids of asthmatic patients, suggesting the important role of S1P in MC-dependent allergic disorders [24,25].

Inhibition of SPHK1 resulted in attenuated allergic airway hyperresponsiveness (AHR) in a model of allergic asthma [26]. Sphingosine signaling promotes cellular interactions in allergic airway inflammation [27]. Exogenous S1P can cause an increased contraction of the bronchi and increased airway resistance, as well as MC and eosinophil recruitment in the lungs in a murine model of allergic airway inflammation [28,29]. These reports suggest the role of sphingolipids in allergic inflammation such as anaphylaxis and asthma.

## 2. Sphingosine-1-Phosphate Receptors

Many of the S1P actions are mediated by five members of the G protein-coupled S1P receptors (S1PR1-S1PR5) with overlapping but distinct couplings to heterotrimeric G proteins [30,31]. After coupling to the G proteins, these receptors can either activate or inhibit downstream signaling pathways, including c-Jun N-terminal kinase (JNK), extracellular signal-regulated kinase (ERK), phosphatidylinositol 3-kinase (PI3K), phospholipase C (PLC), phospholipase D (PLD), signal transducer and activator of transcription 3 (STAT3), Rho, Rac, and cyclic AMP.

S1PR1 can recruit macrophages and MCs to the inflammation sites for tissue repair [32]. S1PR2 regulates the phagocytic capacity [32], while S1PR1 binds to the trimeric G protein Gi/o and activates ERK, Akt, G protein Rac, and phospholipase C (PLC) [33]. S1PR2 mainly activates the low-molecular-wight G protein Rho via G12/13, and Rho activates Rho kinase [34]. S1PR2 mediates allergic asthma by promoting autophagy via Rac1 activation [35]. S1PR2 is also necessary for airway infiltration of the T cells in a mouse model of allergic airway inflammation [36,37]. S1PR3 mainly activates PLC by conjugating Gq, causing Ca^2+^ mobilization, and activating protein kinase C (PKC) [38]. The MCs release S1P that can mobilize P-selectin through S1PR3 [39]. S1PR4 and S1PR5 are present in various immune cells [40]. S1PR4 can promote recruitment of macrophages in a mouse model of psoriasis by increasing monocyte chemoattractant protein 1 (MCP1) production [41]. S1PR5 is critical for migration of natural killer (NK) cells toward S1P [42,43]. The above reports show the roles of S1PRs in allergic inflammation. Thus, the inhibition of S1PRs signaling may provide clues to suppressing various allergic inflammatory diseases. Table 1 shows the locations and functions/mechanism of various S1PRs as well as the cells that express S1PRs.

## 3. Sphingolipids in Atopic Dermatitis

Alteration of the ceramide composition of the epidermis contributes to the pathogenesis of atopic dermatitis (AD) [49,50]. AD is closely associated with the increased serum levels of S1P [51]. Food allergies (FAs) are often accompanied by decreased levels of sphingolipids, including sphingomyelins and ceramides, in the plasma [52]. The number of ceramides is significantly lower in AD patients with FAs than in those without FAs [52]. An increase in the expression of sphingomyelin deacylase and a reduction in the expression of SMase have been reported in AD, which may decrease the total ceramides [53]. High levels of ceramide synthase 4 (CERS4) in AD can enhance the synthesis of short-chain ceramides and confers resistance to bacterial aggressions [54]. These reports indicate the possible role of sphingolipids in AD.

Decreased levels of S1P can result in the growth of Pseudomonas aeruginosa and Staphylococcus aureus, which are known to contribute to the pathogenesis of AD [55]. S1PR2 knockout mice show higher amounts of transepidermal water losses with compromised skin-barrier functions and increased bacteria permeabilities [56]. In response to endoplasmic reticulum (ER) stress, S1P can upregulate cathelicidin, an antimicrobial peptide produced by keratinocytes, by activating NF-κB via binding to the tumor necrosis factor receptor associated factor (TRAF). S1P reduction also compromises the innate immune responses of epithelial cells and macrophages [57].

JTE-013, a selective antagonist of S1PR2, has been shown to suppress 2,4-dinitrochlorobenzene (DNCB)-induced atopic responses in a mouse model. JTE-013 administration can also significantly decrease the levels of pro-inflammatory cytokines (IL-4, IL-13, IL-17, and IFN-γ) in the ears and lymph nodes as well as the levels of chemokines cc ligand 17 (CCL17) and CCL22 in the ears [58]. Deficiency of SPHK1 in the MCs can mitigate ovalbumin (OVA)-mediated remodeling and MC activation in a mouse model of eczema [59]. These reports indicate that S1P signaling can induce or suppress the development of allergic skin diseases such as AD and atopic eczema in a context-dependent manner. Hence, it is probable that the sphingolipids may regulate allergic inflammation, such as asthma.

## 4. Asthma and Sphingolipids

The ER and unfolded protein response (UPR) signals mediate the secretion of bioactive metabolites by the MCs [60,61]. Genome-wide association studies (GWAS) have identified the orosomucoid-like 3 (*ORMDL3*) gene as that is responsible for childhood-onset asthma risk [62,63]. ORMDL3, an ER-resident transmembrane protein, regulates the activity of serine palmitoyltransferase (SPT), a rate-limiting enzyme in the biosynthesis of sphingolipids [64], ER stress, and unfolded protein response (UPR) [65,66], implying the role of *ORMDL3* in asthma.

*ORMDL3* expression has been found to be lower in antigen-activated MCs and can suppress MC activation [67]. Three ORMDL proteins (ORMDL1, ORMDL2, and ORMDL3) can inhibit de novo synthesis of sphingolipids. Polymorphisms of *ORMDL1* and *ORMDL2* have been found to be associated with childhood asthma [68]. The double knockout of *ORMDL2* and *ORMDL3* exhibits increased intracellular levels of S1P in the MCs [69]. Transgenic mice with increased expression of *ORMDL3* showed a reduced level of S1P [70]. Targeting the human ORMDL3 protein led to increased airway remodeling (smooth muscle contraction, fibrosis, mucous production) and an enhanced IgE responses compared to the wild-type mice following an allergen challenge [71]. These reports suggest the regulatory role of *ORMDL3* in allergic inflammation in relation to S1P.

The S1P levels were observed to be higher in the BAL fluids of patients with asthma [48]. CYM50358, a selective antagonist of S1PR4, can inhibit the increase in eosinophils and lymphocytes in the bronchoalveolar lavage fluids [48]; its administration can also inhibit the increase of IL-4 cytokines and serum IgE levels [48]. S1P can disrupt the epithelial cell barrier integrity (tight junctions) in the respiratory system to induce allergic airway inflammation [72,73]. The S1P/SPHK/S1PRs pathway can trigger airway hyperresponsiveness (AHR) in ovalbumin (OVA)-sensitized mice [74]. Muscarinic receptor (MR) signaling has been implicated in the development of asthma [75], and its downstream signaling can lead to constriction of the peripheral airways via activation of SPHKs and the release of intracellular Ca2+ levels [76]. Systemic administration of S1P has been shown to increase the airway resistance and cholinergic activity in a whole mouse lung model [77]. These reports suggest the role of S1P signaling in allergic inflammation, such as asthma.

In a murine model of chronic asthma, SK1-I, an inhibitor of SPHK1, has been shown to decrease eosinophil numbers as well as levels of cytokines (such as IL-4, IL-5, IL-6, IL-13, IFN-γ, and TNF-α) and chemokines (such as eotaxin, and CCL2) in bronchoalveolar lavage fluids (BAL) fluids [26]. Treatment with an SPHK inhibitor can hence improve immune responses in the mouse model of asthma [78]. Inhibition of SPHKs by SK1-II in OVA-sensitized mice can abrogate epithelial to mesenchymal transition (EMT), pulmonary TGF-β upregulation, fibroblasts recruitment, and AHR [79]. S1P-mediated effects can also be abrogated by JTE-013 (an S1PR2 antagonist) or Y-27632 (an inhibitor of Rho kinase), indicating the role of S1PR2 in allergic airway inflammation [46,47]. JTE-013 can decrease the inflammatory cell infiltration and goblet cell production in asthmatic mice tissues [35]. These reports indicate that S1P/S1PRs signaling can contribute to the pathogenesis of asthma.

## 5. Anaphylaxis and Sphingolipids

Ceramide and S1P can regulate a diverse range of cellular processes concerning immunity and inflammation [80,81]. S1P acts as a signaling component within the MCs [82]. This implies the role of S1P in mast cell activation during allergic inflammation.

FcεRI is necessary for passive cutaneous anaphylaxis (PCA) reaction by activating MAPK signaling [83]. PCA is accompanied by ear swelling, enhanced vascular permeability, and angiogenesis. The allergen binds to at least two molecules of IgE (Figure 2), and IgE then binds to the alpha subunit of FcεRI, resulting in cross linking of FcεRI (Figure 2). FcεRI then activates ERK, p38 mitogen-activated kinase (p38 MAPK), JNK, and PI3K [83,84,85] (Figure 2), in addition to increasing the expression of Th2 cytokines [86,87] (Figure 2). FcεRI signaling can activate SPHK1, subsequently leading to S1P production in rat basophilic leukemia 2H3 (RBL2H3) cells [88]. It is probable that Th2 cytokines could activate SPHK1 and increase the level of S1P in allergic inflammation.

Allergen (antigen) binding to IgE can activate FcεRI signaling, and IgE-bound FcεRI can activate Src kinases, such as Lyn and Fyn, which in turn can activate PLCγ to convert phosphatidylinositol 4,5-bisphosphate (PIP2) into inositol 1,4,5-triphosphate (IP3) and diacylglycerol (DAG). IP3 can increase intracellular calcium concentration by acting through the IP3 receptor on the membrane of the ER. Calcium ions can induce degranulation of MCs, increasing the secretion of histamine and heparin. DAG and calcium ions can activate PKC, which in turn activates P38 mitogen-activated protein kinase MAPK (P38MAPK) as well as increases the expression levels of B cell lymphoma/leukemia 10 (BCL10) and mucosa-associated lymphoid tissue lymphoma translocation protein 1(MALT1). ERK, JNK, and P38 MAPK can activate nuclear factor-κB (NF-kB) to increase the expression levels of T helper 2 (Th2) cytokines. PI3K, which is activated by Lyn and Fyn, can convert PIP2 into phosphatidyl inositol 3,4,5-triphosphate (PIP3), which can activate JNK through RAC and mitogen-activated protein kinase 4 (MAPK4). PI3K can also activate ERK1/2, which in turn can activate phospholipase A2 (PLA2) and increases secretion of lipid mediators, such as leukotrienes and prostaglandins. The prostaglandins may then recruit effector cells, and the leukotrienes can increase vascular permeability. JNK, P38MAPK, and ERK1/2 can increase the production of cytokines in allergen (antigen)-stimulated MCs. Lipid mediators and cytokines also induce MC activation. ↑ denotes increased level/activity; The black and dotted arrows denote the directions of reaction; α, β, and γ denote the subunits of FcεRI; the black stars denote the potential targets of the reactive oxygen species (ROS). MEK, mitogen-activated protein kinase.

IgE-treated MCs showed increased levels of S1P and sphingosine-1-phopsphate receptor 3 (S1PR3) [89]. S1P is also necessary for calcium influx to activate PKC for MC granulation in the mouse model of passive systemic anaphylaxis (PSA) [90]. Mice lacking both the isoforms of SPHKs with undetectable levels of circulating S1P show impaired survival after anaphylactic reaction [91]. These reports further suggest the role of S1P in allergic inflammation, such as anaphylaxis.

Activation of P42/P44 MAPK, Akt, and nuclear factor-κB (NF-kB) pathways is essential for S1P-induced COX-2 gene expression [92]. S1PR1/3 activated by S1P can stimulate epidermal growth factor receptor (EGFR)/PI3K/Akt/MAPKs/AP-1 signaling to upregulate COX-2 [44]. The S1P/SPHK1 pathway can mediate colon carcinogenesis by regulating COX-2 expression and prostaglandin E2 (PGE2) production [93]. Activation of S1PR3 by S1P can enhance the inflammatory and metastatic potential of breast cancer cells by inducing COX-2 expression and PGE2 signaling. [94]. COX-2 is necessary for PSA as well as for enhanced metastatic potential of B16F1 melanoma cells by PSA [95]. It is thus reasonable that S1P may increase COX-2 expression in the MCs to mediate allergic inflammation. Metformin can inhibit FcεRI- and aryl hydrocarbon receptor (AhR)-mediated PCA in vivo [96]. Metformin inhibits FcεRI-mediated degranulation, IL-13, and S1P secretion in bone marrow-derived mast cells (BMMCs) [96]. These reports suggest that the crosstalk between FcεRI signaling and S1P signaling can mediate allergic inflammation, such as anaphylaxis. Figure 3 shows that the activation of FcεRI signaling increases S1P production, which in turn activates FcεRI signaling and induce mast cell degranulation. Table 2 shows the roles and functions of sphingolipids in AD, asthma, and anaphylaxis.

## 6. FcεRI/S1P/HDACs in Anaphylaxis

Anaphylaxis is a severe and life-threatening multisystem syndrome that necessitates immediate medical attention [97]. MCs can mediate various allergic reactions by secreting pro-inflammatory mediators [98,99]. It is thus probable that FcεRI signaling and downstream molecules may mediate anaphylaxis.

Histone deacetylases (HDACs) remove acetyl groups from histones and tighten the binding of DNA to the histones. The regulatory roles of HDACs in gene expression allow HDACs to act as regulators of allergic inflammation. FcεRI signaling is necessary for the induction of HDAC3 and MCP1 (CCL2) [100], which are in turn necessary for PCA [100]. HDAC3 directly increases MCP1 expression in antigen-stimulated RBL2H3 cells [100]. The recombinant MCP1 protein enhances vascular permeability and angiogenic potential [100], suggesting that MCP1 can mediate PCA. We previously reported that PSA enhanced the tumorigenic and metastatic potential of mouse melanoma cells through the induction of HDAC3, MCP1, and CD11b (a macrophage marker) expressions [101]. MCP1 mediates the cellular interactions that promote enhancement of the metastatic potential of cancer cells by PSA [101]. miR-384 targets HDAC3 and suppresses the positive feedback relationship between anaphylaxis and the anaphylaxis-enhanced metastatic potential of cancer cells [101]. It is probable that extracellular MCP1 may promote these cellular interactions. Thus, the HDAC3–MCP1 axis mediates anaphylaxis-enhanced tumorigenic and metastatic potential by promoting cellular interactions.

Saturated fat-induced NF-κB signaling and elevated expressions of TNFα and MCP1 mRNA in HepG2 cells can be blocked by targeted knockdown of S1PR1 [102]. Since MCP1, increased by FcεRI signaling, is necessary for allergic inflammation both in vitro and in vivo, it is reasonable to hypothesize that S1PR signaling may be involved in allergic inflammation, such as anaphylaxis, by regulating the expression of MCP1. Thus, the crosstalk between FcεRI signaling and S1PR signaling may mediate allergic inflammation.

Antigen stimulation can induce the expression and activity of transglutaminase (TGase II) by activating NF-kB in RBL2H3 cells [103]. TGase II is responsible for the increased production of ROS as well as the increased expression of prostaglandin E2 synthase (PGE2 synthase) and HDAC3 in RBL2H3 cells [103]. TGase II also mediates PCA [103]. Since HDAC3 and PGE2 synthase can mediate allergic inflammation, it is reasonable to hypothesize that the TGase II–HDAC3 axis may regulate S1P production during allergic inflammation.

Sphingolipid biosynthesis is initiated by SPT. Tubacin, an inhibitor of HDAC6, can inhibit de novo sphingolipid synthesis by suppressing SPT activity [104]. This suggests the possible role of HDAC 6 in S1P production. This also suggests the possible role of HDAC6 in anaphylaxis. Antigen stimulation can also induce the expression of HDAC6 in RBL2H3 cells. HDAC6 is necessary for the increased expression of HDAC3 in antigen-stimulated RBL2H3 cells [105]. HDAC6 is necessary for the enhanced tumorigenic potential of melanoma cells by PSA [105]. HDAC6 promotes cellular interactions by increasing IL-27 during anaphylaxis [105]. Thus, the HDAC6–IL-27 axis may increase S1P production during allergic inflammation.

PSA can enhance the tumorigenic potential of cancer cells by promoting cellular interactions involving cancer cells, MCs, macrophages, and various stromal cells [106]. This implies the role of S1P signaling in cellular interactions during the anaphylaxis-promoted enhanced tumorigenic potential of cancer cells. S1P promotes M2 macrophages polarization and contributes to the pathogenesis of endometriosis [107]. M2 macrophages polarization and S1P signaling contribute to the pathogenesis of colitis-associated colon cancer [108]. We previously reported that M2 macrophages polarization induced by cellular interactions mediated anaphylaxis and anaphylaxis-promoted enhanced the tumorigenic and metastatic potential of melanoma cells [101]. M2 macrophages polarization was induced by antigen-stimulated mast cells [101]. These reports suggest that S1PRs signaling may mediate anaphylaxis by promoting cellular interactions. Considered together, these reports suggest that FcεRI/S1P/HDACs signaling can mediate anaphylaxis by promoting cellular interactions involving MCs and macrophages.

## 7. S1P and HDAC Activity

Inhibition of HDAC3 can diminish S1P production in osteoclasts [109]. S1P treatment might ameliorate cardiac hypertrophic responses, which are partly mediated by the suppression of HDAC2 activity and the upregulation of KLF4 [110]. S1P can target HDACs and TRAF2 to mediate various cellular processes [111]. These reports suggest a positive feedback relationship between S1P level and HDAC activity. The SPHK1/2 inhibitors can promote the activity of HDAC1 and inhibit the histone acetylation of the Krüppel-like factor 4 (KLF4) promoter regions to regulate M1 to M2 microglial polarization [112]. These reports suggest that S1P signaling mediates cellular interactions during allergic inflammation by regulating HDAC activity.

S1P is metabolized to ∆2-hexadecenal (∆2-HDE) and ethanolamine phosphate by S1P lyase (S1PL) in mammalian cells (Figure 1). The addition of exogenous ∆2-HDE to lung epithelial cell nuclear preparations can inhibit HDAC1/2 activities and can increase acetylation of histones H3 and H4 [113]. S1PL deficiency is associated with reduced HDAC activity and down-regulation of HDAC isoenzymes [114]. HDAC6 controls the acetylation of several cytosolic proteins and most prominently tubulin [45,115]. The use of selective inhibitors of HDAC6 is under investigation as a potential treatment strategy for inflammatory diseases owing to their ability to regulate SPT, inflammatory cells, and cytokines [116]. It is probable that HDAC6 mediates anaphylaxis by activating S1P signaling.

Figure 4 shows a proposed model of allergic inflammation in relation to the crosstalk between FcεRI and S1P/S1PR signaling. S1P generated by activated FcεRI signaling plays a critical role in PCA/PSA/TpCR by regulating the expressions and/or activities of HDACs, such as HDAC2 and HDAC3 (Figure 4). These reports indicate that crosstalk between FcεRI and S1P/SPHK/S1PR signaling may regulate anaphylaxis through its effects on HDACs.

The IgE-bound FcεRI can increase the expression of HDAC3, which can bind to Rac1, and decrease the expression of HDAC2 by inducing ubiquitination of HDAC2. HDAC3 can also bind to the promoter sequences of MCP1 to increase its expression. HDAC3 and MCP1 can mediate PCA, PSA, and triphasic cutaneous reactions (TpCRs). In the absence of antigen stimulation, HDAC2 can bind to the promoter sequences of MCP1 to decrease the expression of MCP1. S1P levels can be increased by SPHKs activated by FcεRI signaling. S1P can inhibit HDAC2 activity by increasing HDAC3 activity, which in turn can increase the expression level of MCP1 to mediate allergic inflammation, such as PCA, PSA, and TpCR.

## 8. Mast Cells and Sphingosine Kinases

MCs express both SPHK1 and SPHK2 isoforms, which are activated following FcεRI stimulation, and S1P generated by the SPHKs can mediate the IgE-promoted release of mediators and cytokines [117,118,119,120].

Src kinases, such as Lyn and Fyn, can interact with and activate SPHK1 in the MCs to promote the recruitment of SPHK1 to FcεRI [121]. Deficiency of Fyn is known to ablate SPHK activation [121,122]. RNAi and genetic deletion experiments have suggested the critical role of SPHKs in MC function, indicating that SPHK1 and SPHK2 may show some redundancy [123].

Calcium influx is critical for FcεRI-mediated MC degranulation and cytokine generation [124]. Inhibition of SPHK activity or short hairpin-mediated SPHK1 silencing can block calcium flux and decreases IL-33 expression induced by IgE/Ag activation [125]. These reports suggest that activation of SPHKs by FcεRI signaling is necessary for MC activation in allergic inflammation.

## 9. Mast Cells and Sphingosine Receptors

SPHK1 knockout mice showed reduced plasma S1P levels [126]. FcεRI-mediated activation of SPHKs can generate extracellular S1P, and the ATP-binding cassette (ABC) superfamily of transporters can transport S1P [127] (Figure 3).

MCs express S1PRs [128,129], and FcεRI-mediated activation of SPHKs can induce trans-activation of S1PR1 and S1PR2, resulting in MC degranulation [128].

The S1P-S1PR signaling pathway is critical for both MC degranulation and migration [130]. The activation of S1PR1 can promote migration of MCs toward antigens, while the activation of S1PR2 can trigger MC degranulation [131]. S1PR2 antagonist JTE-013 was shown to attenuate severe hypothermia and reduce serum histamine levels in a mouse model of anaphylaxis [132]. S1PR3 signaling promoted pulmonary inflammation and fibrosis via connective tissue growth factor (CTGF) expression [133]. This implies the role of S1PR3 in allergic inflammation, such as asthma. Thus, S1PRs promote allergic inflammation by mediating MC activation. The above reports suggest that S1PRs can serve as targets for developing anti-allergy drugs.

## 10. MicroRNAs and S1P Signaling

MicroRNAs (miRNAs/miRs) are noncoding single-stranded RNAs of 18–24 nucleotides in length. Since microRNAs target multiple genes, they are involved in various life processes such as cellular interactions [106], cell proliferation [134], and inflammation [135,136].

miR-126 can promote IgE-mediated MC degranulation associated with the PI3K/Akt signaling pathway by promoting Ca2+ influx [137]. S1P can upregulate the expression of MCP1 and decrease miR-1249-5p expression in macrophages [138]. miR-1249-5p can bind to the 3′-UTR of MCP1 and suppress the expression of MCP1 in S1P-treated macrophages [138]. Thus, miR-1249-5p can negatively regulate allergic inflammation. Thus, MCP1 may enhance levels of S1P during allergic inflammation. High level of miR-221 has been reported in a murine model of lung asthma [139]. miR-221 can increase total cells and eosinophil numbers in a murine model of asthma and stimulate IL-4 secretion in MCs via PTEN, p38, and NF-κB pathways [140]. miR-21 is also upregulated in multiple asthmatic mouse models induced by house dust mite, OVA, or Aspergillus fumigatus [141]. These reports suggest the roles of miRNAs in allergic inflammation.

miRNAs can target components of the S1P signaling pathway (Table 3). For example, miR-506 can target SPHK1 [142,143] and suppress proliferation of airway smooth muscle cells by inducing apoptosis and inhibiting MCP1 expression and NF-κB activity [144]. miR-124 is decreased in patients with atopic eczema and miR-124 can suppress atopic eczema [145]. miR-125b-1-3p, miR-133b, and miR-363 target S1PR1 [146,147]. miR-130a-3p and miR-613 target SPHK2 [148,149], whereas miR-125b [150] target S1PL mRNA. miR-130a-3p can suppress allergic asthma by inhibiting M2 macrophages polarization [151]. Overexpression of miR-125b-1-3p has been found in the sera of patients with asthma [152], and also in patients with allergic rhinitis [153]. miR-133b mimic was shown to attenuate allergic rhinitis by decreasing the levels of Th2 cytokines and NLR family pyrin domain containing 3 (Nlrp3) [154]. These miRNAs may contribute to the pathogenesis of allergic inflammation by regulating the expression levels of components of S1P signaling pathways.

Table 3 shows the functions of miRNAs that target S1P/SPHK/S1PRs signaling in allergic inflammation. Figure 5A shows the miRNAs that target the components of S1P/SPHK/S1PRs signaling. Figure 5B shows the potential binding sites for various transcription factors in the promoter sequences of S1PRs. The roles of these transcription factors in the expression regulation of S1PRs or allergic inflammation have not been investigated extensively. TargetScan analysis revealed miRNAs that can bind to the 3′ untranslated regions (UTRs) of S1PRs (Figure 5C). The roles of these miRNAs in the expression regulation of S1PRs or allergic inflammation have not been investigated extensively. Thus, miRNAs that can target S1P signaling components can be developed as anti-allergy drugs.

## 11. Conclusions

It is known that IgE-bound MCs activate FcεRI signaling, which in turn leads to the activation of ERK, P38 MAPK, JNK, and PI3K. Since FcεRI signaling increases S1P production in MCs, it would be necessary to examine the effects of ERK, P38 MAPK, JNK, and PI3K on S1P production in allergic inflammation both in vitro and in vivo. Since FcεRI signaling also increases the production of Th2 cytokines, it would be interesting to examine the effects of Th2 cytokines on S1P production in allergic inflammation.

Based on previous reports, sphingolipid signaling is becoming an important target for developing anti-allergy drugs. It is thus necessary to examine whether S1P can induce the hallmarks of allergic inflammation, such as PCA and PSA, in an IgE-independent manner. Moreover, it may be necessary to identify the S1P-regulated genes, which can provide clues to the understanding of allergic inflammation mediated by crosstalk between FcεRI signaling and S1P signaling.

Since S1PR2 is involved in allergic inflammation, the identification of the downstream targets of S1PR2 is necessary for developing anti-allergy drugs. RNA sequencing can identify the downstream targets of S1PR2, which may include components of FcεRI signaling. Proteins that can bind to S1PRs proteins can also serve as targets for developing anti-allergy drugs. It is hence necessary to identify the full-length protein structures of S1PRs, which can enable screening of the inhibitors that regulate the expression and/or activity of S1PRs.

Since SPHKs can mediate allergic inflammation, it may be necessary to screen inhibitors of SPHKs based on the full-length protein structures of SPHKs. It is also necessary to identify genes that may be regulated by these inhibitors. These inhibitors can be further developed as anti-allergy drugs and could suppress FcεRI signaling to prevent allergens (antigens) from increasing the production of Th2 cytokines and inducing MC degranulation. It is probable that these inhibitors may also suppress in vivo allergic inflammation, such as anaphylaxis and asthma.

Allergic inflammation involves cellular interactions among the MCs, macrophages, and endothelial cells, which are mediated by exosomes. It will be interesting to examine the presence of the components of S1P/S1PR signaling in the exosomes. It is also necessary to examine whether the components of S1P signaling could mediate cellular interactions in allergic inflammation. It is probable that S1P phosphatase and/or S1PL may inhibit allergic inflammation by negative regulation of FcεRI signaling and cellular interactions.

MiRNAs are known to regulate expression levels of the components of S1P/S1PR signaling, such as SPHKs and S1PRs [146,147]. Thus, miRNA mimics and miRNA inhibitors can be developed as anti-allergy drugs. However, only a few miRNA mimics or miRNA inhibitors are being investigated in clinical trials. miRNAs that target the components of S1P/S1PR signaling have not been tested in clinical trials concerning allergic diseases. Unlike siRNAs, miRNAs have multiple targets; this property may cause off-target effects. Modifications of the miRNA mimics or miRNA inhibitors will therefore be necessary for improving the stability, pharmacokinetics, and pharmacodynamics of miRNA-based therapeutics. It is also necessary to understand the roles of miRNAs targeting the components of S1P/S1PR signaling in allergic inflammation before developing miRNA-based anti-allergy drugs.

Complete understanding of the crosstalk between FcεRI signaling and S1P signaling in allergic inflammation will thus be necessary for developing anti-allergy drugs in the future. Given the fact that anaphylaxis can enhance the tumorigenic potential of cancer cells, it is expected that the anti-allergy drugs can also be used as anti-cancer drugs

## Figures and Tables

**Figure 1 ijms-23-13892-f001:**
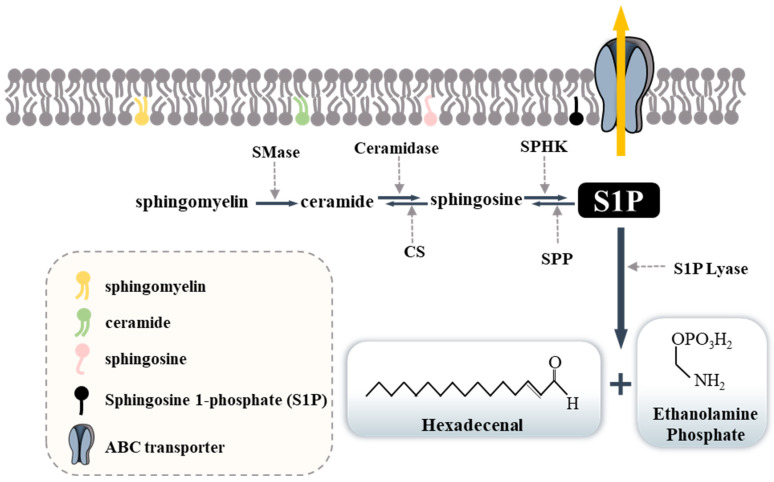
Metabolic pathway of sphingolipids. Sphingomyelin is present in the cell membrane and can be converted to ceramide by sphingomyelinase. Ceramidase can the convert ceramide into sphingosine, which is phosphorylated by the SPHKs and converted into S1P. Sphingosine can be converted into ceramide by ceramide synthase. S1P can be degraded into hexadecenal and ethanolamine phosphate by S1P lyase. S1P is also be subjected to extracellular transport by ABC transporters. S1P can be converted into sphingosine by sphingosine phosphate phosphatase. The arrows (black, yellow, dark yellow, and dotted) denote the direction of reactions. CS, ceramide synthase; SPP, sphingosine-1-phosphate phosphatase.

**Figure 2 ijms-23-13892-f002:**
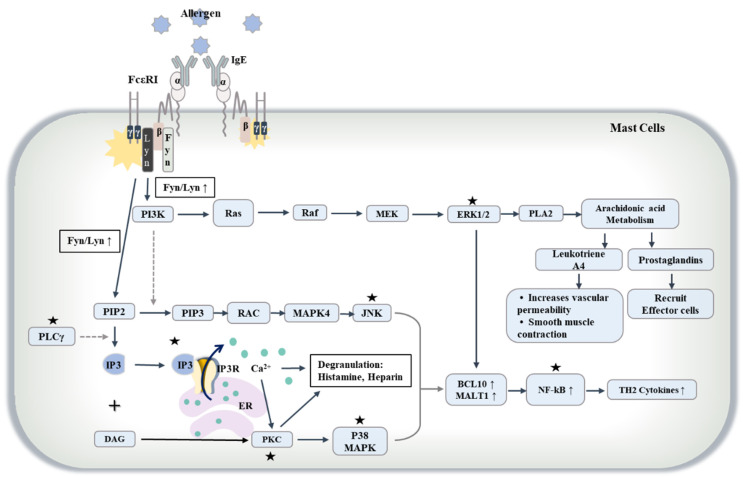
FcεRI signaling in allergic inflammation.

**Figure 3 ijms-23-13892-f003:**
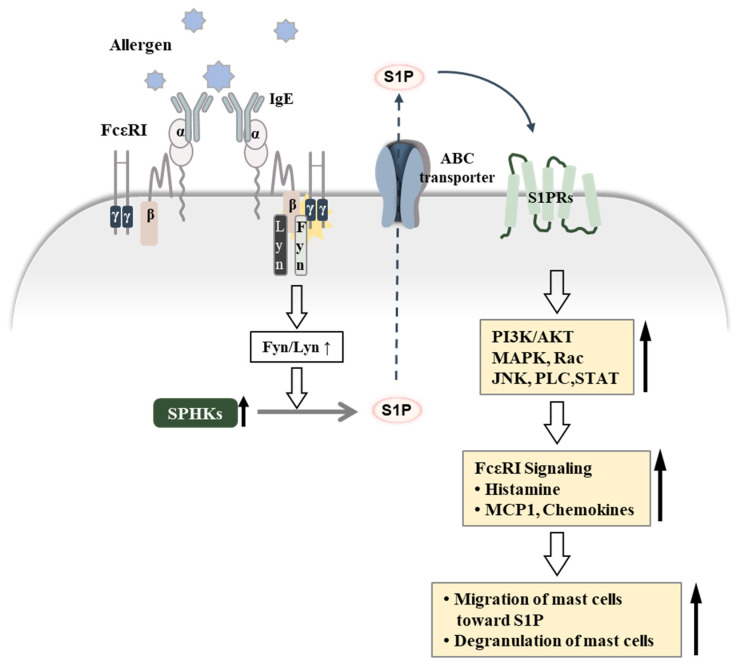
Crosstalk between FcεRI and S1PRs in allergic inflammation. Allergens can be bound by two molecules of IgE, resulting in dimerization of FcεRI. IgE-bound FcεRI can activate SPHKs, which in turn increase the levels of S1P. S1P undergoes extracellular transport by the ABC transporter. Extracellular S1P can bind to S1PRs, which in turn can regulate multiple signaling pathways to activate FcεRI signaling and are responsible for degranulation in MCs. The increased calcium ions by FcεRI signaling are known to induce MC granulation by increasing the secretion of histamine. Here, ↑ denotes increased level/activity, the hollow and dotted arrows denote directions of reaction, and α, β, and γ denote the subunits of FcεRI.

**Figure 4 ijms-23-13892-f004:**
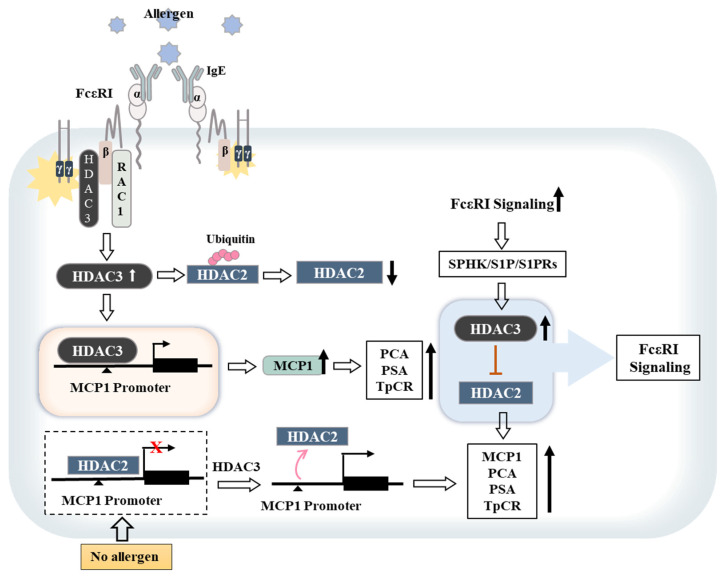
Proposed model of FcεRI-promoted allergic inflammation in relation to HDAC2/3 and S1P signaling. The hollow arrows denote the direction of reactions; the T bar arrows denote negative regulation; ↑ denotes increased level/activity; ↓ denotes decreased level/activity; Red X denotes transcription repression. Ub, ubiquitination.

**Figure 5 ijms-23-13892-f005:**
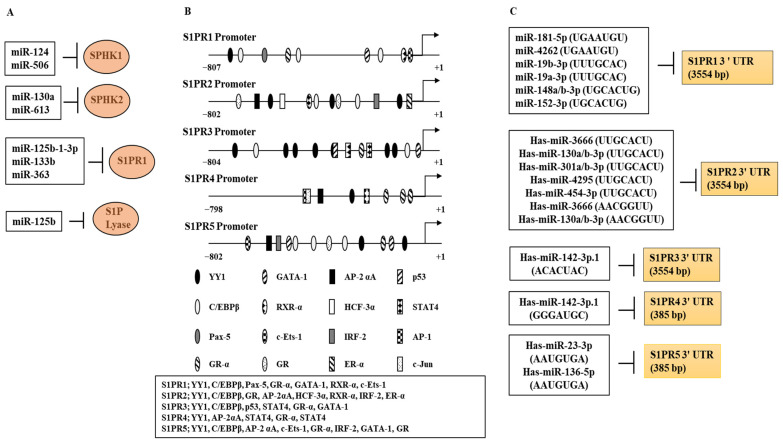
Expression regulation of SPHK and S1PRs. (**A**) microRNAs that target components of S1P signaling are shown. The T bar arrows denote negative regulation. These miRNAs directly decrease the expression of SPHK1, SPHK2, S1PR1, or S1P lyases. SPHK, sphingosine kinase: S1PR1, sphingosine-1-phosphate receptor 1. (**B**) Potential binding sites for transcription factors in the promoter sequences of S1PRs. Here, +1 denotes the transcription start site. C/EBP, CCAAT box enhancer binding protein; ER, estrogen receptor; GATA-1, GATA binding protein 1; GR, glucocorticoid receptor; HCF, host cell factor; IFR-2, interferon regulatory factor 2; RXR, retinoid X receptor; STAT4, signal transducer and activator of transcription 4; YY1, yin yang 1. (**C**) TargetScan analysis reveals miRNAs that can bind to the 3′ untranslated regions (UTR) of S1PRs. Sequences in the parentheses denote sequences of miRNAs that can form base pairs with the 3′ UTRs of S1PRs. The roles of these miRNAs in allergic inflammation or the expression regulation of S1PRs have not been studied yet.

**Table 1 ijms-23-13892-t001:** Locations and roles of S1PRs in allergic inflammation.

Receptor	Location/Cells	Functions/Mechanisms	Refs
S1PR1	Endosome Plasma membrane B cells, Macrophages, MC, Dendritic cells	Recruits mast cells and macrophages for tissue repair COX-2 ↑ Anti-inflammatory cytokines ↓ FcεRI signaling ↑	[32,43,44,45]
S1PR2	Plasma membrane B cells, Macrophages, MC, Dendritic cells	Phagocytic activity Airway infiltration of T cells Inflammatory cell filtration and goblet cell production in asthma FcεRI signaling and mast cell degranulation	[32,35,36,37,46,47]
S1PR3	Plasma membrane B cells, Macrophages, MC, Neutrophils	P-selectin-dependent leukocyte rolling PLC activity ↑	[38,39]
S1PR4	Plasma membrane Mitochondria B cells, T cells, Macrophages, MC, Eosinophils	Recruits macrophages in a mouse model of psoriasis Eosinophils ↑ Lymphocytes in BAL fluids ↑ IL4 ↑ Serum IgE ↑	[41,48]
S1PR5	Plasma membrane NK cells, MC, Eosinophils	Migration of NK cells toward sphingosine-1- phosphate	[42,43]

↓ denotes decreased level/activity; ↑ denotes increased level/activity; BAL fluids, bronchoalveolar lavage fluids; COX-2, cyclooxygenase; IL-4, interleukin-4; NK, natural killer.

**Table 2 ijms-23-13892-t002:** Roles of sphingolipids in allergic diseases.

Disease	Markers (Sphingolipids)	Functions/Mechanisms	Refs
Atopic dermatitis/allergic skin inflammation	S1P ↑ Ceramide ↓ Sphingomyelinase ↓	- Decreased level of ceramide leads to AD. - Knock-out of S1PR2 leads to trans - epidermal water loss and AD. - High-level of S1P can cause AD. - Inactivation of S1PR2 suppresses MC activation in eczema	[52,56,57,59]
Asthma	S1P ↑	- S1P/SPHK signaling enhances airway remodeling. - ORMDL inhibits allergic asthma by decreasing S1P level - S1P promotes allergic asthma by recruiting eosinophils through chemokines - IgE and T cells mediate S1P-promoted allergic asthma - S1PR2 mediate S1P-promoted allergic asthma	[24,25,36,37,67,71,74]
Anaphylaxis	S1P ↑ Sphingosine ↓	- S1P/S1PR signaling promotes MC degranulation. - S1PR2 is critical for PSA. - S1P enhances production of leukotrienes and TNF-α. - S1P acts a ligand for S1PRs and intracellular second messenger.	[36,37,90,96]

↓ denotes decreased level/activity; ↑ denotes increased level/activity.

**Table 3 ijms-23-13892-t003:** Roles of miRNAs targeting S1P signaling in allergic inflammation.

microRNAs	Target	Functions/Mechanisms in Allergic Inflammation	Refs
miR-506	SPHK1	Suppresses proliferation of airway smooth muscle cells by inducing apoptosis MCP1 ↓ NF-κB activity ↓	[142,143,144]
miR-124	SPHK1	Suppresses atopic eczema by decreasing levels of IL-8, CCL5, and CCL8	[145]
miR-363	S1PR1	Not known in allergic inflammation Suppresses proliferation of hepatic cancer cells	[147]
miR-130a-3p	SPHK2	Suppresses allergic asthma by inhibiting M2 macrophages polarization	[148]
miR-613	SPHK2	Not known	[149]
miR-125b-1-3p	S1PR1	↑ in the sera of patients with asthma	[152]
miR-125b	S1P lyase	↑ in the sera of patients with allergic rhinitis	[153]
miR-133b	S1PR1	Suppresses allergic rhinitis by decreasing the levels of Th2 cytokines	[154]

↓ denotes decreased level/activity; ↑ denotes increased level/activity; CCL5, cc chemokine ligand 5; IL-8, interleukin-8.

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
