# Peer review of "The Crosstalk between FcεRI and Sphingosine Signaling in Allergic Inflammation"

_ijms, 2022, doi:10.3390/ijms232213892_

Round 1

Reviewer 1 Report

In this Review article, the authors describe the relationship between the IgE receptor FcεRI-mediated signaling and sphingosine signaling in allergic inflammatory. Although it was previously known that FcεRI-mediated activation of sphingosine kinases and production of sphingosine-1-phosphate lead to mast cell activation, this article contains the latest findings on the field and may provide useful information to readers. I have a few suggestions to improve this article.

1.     Throughout the text: There are a mixture of various notations such as S1P, sphingosine-1-phosphate, and sphingosine-1-phosphate (S1P). Abbreviations should be spelled out upon first usage. After that, please keep the notation consistent throughout the text. The same is true for sphingosine kinase (SPHK), rat basophilic leukemia 2H3 (RBL2H3), and mast cell (MC).

2.     Throughout the text: Monocyte chemoattractant protein 1 (MCP1) is mentioned frequently in this article. Thus, the role of MCP1 in allergic inflammation should be explained in detail.

3.     Table 1: I would suggest that Table 1 include information on the cell types expressing these receptors. In the text, the authors describe that S1PR4 and 5 are localized to lymphatic and nervous cells (page 3, line 90).

4.     Legend to Figure 2 (page 6, line 185): In this context, “mast cell activation” is more appropriate than “mast cell degranulation”.

5.     Figure 3: Please illustrate in more detail the mechanism by which FcεRI-mediated signaling activates SPHKs. In the text, the authors describe that Fyn kinase can interact with SPHK1 and SPHK2 proteins (page 10, line 344).

Author Response

Dear Sir

Thanks for excellent suggestions. I made changes according to the suggestions.
In this revision, I sought help for English problems. I also send English certificate.

In this Review article, the authors describe the relationship between the IgE receptor FcεRI-mediated signaling and sphingosine signaling in allergic inflammatory. Although it was previously known that FcεRI-mediated activation of sphingosine kinases and production of sphingosine-1-phosphate lead to mast cell activation, this article contains the latest findings on the field and may provide useful information to readers. I have a few suggestions to improve this article.

Q1. Throughout the text: There are a mixture of various notations such as S1P, sphingosine-1-phosphate, and sphingosine-1-phosphate (S1P). Abbreviations should be spelled out upon first usage. After that, please keep the notation consistent throughout the text. The same is true for sphingosine kinase (SPHK), rat basophilic leukemia 2H3 (RBL2H3), and mast cell (MC).

Ans. Thanks. I agree. I made changes as you suggested. Please take look at new manuscript.

Q2.     Throughout the text: Monocyte chemoattractant protein 1 (MCP1) is mentioned frequently in this article. Thus, the role of MCP1 in allergic inflammation should be explained in detail.

Ans. Thanks for excellent suggestions. I agree. I describe the role of MCP1 in allergic inflammations: Please take look at lines 80-81, 229-237. We and others previously reported that MCP1 promoted cellular interactions during allergic inflammations.

Q3.     Table 1: I would suggest that Table 1 include information on the cell types expressing these receptors. In the text, the authors describe that S1PR4 and 5 are localized to lymphatic and nervous cells (page 3, line 90).

Ans. I add information on cells that express S1PRS. Please take look at new table 1. I change [S1PR4 and 5 are localized to lymphatic and nervous cells] into S1PR4 and S1PR5 are present in various immune cells [40]. Please take look at new manuscript (lines 78-79).      

Q4.     Legend to Figure 2 (page 6, line 185): In this context, “mast cell activation” is more appropriate than “mast cell degranulation”.

Ans. Thanks. I change it as you suggested. Please take look at new manuscript (line 173). Mast cell is also changed into MC.   

Q5.     Figure 3: Please illustrate in more detail the mechanism by which FcεRI-mediated signaling activates SPHKs. In the text, the authors describe that Fyn kinase can interact with SPHK1 and SPHK2 proteins (page 10, line 344).

Ans. Thanks. I change figure 3. New figure 3 shows receptor crosslinking. Old figure 3 did not show receptor crosslinking correctly. I change it into Src kinases such as Lyn and Fyn can interact with and activate SPHK1 in MCs and promotes the recruitment of SPHK1 to FcεRI [123]. Please take look at new manuscript (lines 311-312). It is probable that the activated Fyn/Lyn by FcεRI may increase phosphorylation of SPHKs and promote recruitment of SPHKs into FcεRI

Reviewer 2 Report

This work provides general information about the sphingosine-1-phosphate and its receptors and its relationship with the signaling system of the high affinity IgE receptor in mast cells.   The text gives a general perspective of SP1 signaling pathways and suggests some cross-talk points with the FceRI receptor. The provided information is relevant since bioactive lipids have emerged as an important alternative to control inflammation. It is important to point out that the article exposes relevant information in the field of miRNAs and HDAC in the S1P system, which is quite novel and useful to the understanding of the long-term and complex actions of this lipid. However, information is not presented in a clear form, some concepts are repeated in distinct paragraphs and it is not possible to fully understand the message of the authors, since there is no discussion of the data. Some  sections are composed of phrases exposing results of distinct papers and no final statement making an analysis or conclusion can be found. Finally, figures regarding the FceRI signal transduction system are quite incomplete and a number of elements should be included.

Specific examples:

In several parts of the review, main findings are mentioned without a clear context or conclusions, and some details on MC activation are duplicated. For example, section about Anaphylaxis and Sphingolipids starts with a very brief description of the signal transduction system of FceRI receptor but this description is not connected with the induction of anaphylaxis. Description of anaphylaxis appears until line 250 and lines 251 to 254 mention the role of pro-inflammatory mediators on the induction of allergic asthma, atopic dermatitis (which was already mentioned on a previous section).

Lines 205 to 210 mention few references in which SIP/SPHK1 axis is explored but there is no conclusion of the presented information, since line 211 only states that “There is a role of S1P in anaphylaxis”

Line 213 mentions some data on the importance of COX2 expression, but data were obtained in cardiac fibroblasts and colon carcinogenesis. Line 217 contains information on the potential metastatic potential of breast cancer cells, but it is not clear the relationship of those data with S1P effects on Mast Cells. 

Data on metformin effects on passive cutaneous anaphylaxis (PCA) mentioned on lines 221-223 are not discussed, since authors only mentions that the cross-talk S1P and FceRI plays a critical role on anaphylaxis.

Main ideas on section 6 are not easily to follow. It starts with reactions mediated by MCs (information that was already mentioned in previous sections), then DNA deacetylases are mentioned but the specific function of those enzymes on allergic reactions is not mentioned or discussed. In line 265, some information about lung tumors is presented but there is no explanation about the relationship between lung tumors, DNA deacetylases and allergies.

Ideas exposed in lines 266 to 271 are not very clear. The experimental model and conditions in which studies were made are quite important here, since no relationship with allergies can be appreciated.

Section 7 contains information already included in previous paragraphs and it is not clear what is the message and, more importantly, what is the conclusion? Detailed analysis and discussion must be included in this section.

Specific comments:

Lines 50 and 51 should be part of Figure 1 Legend.

FceR1 must be written with a the latin number one (I)

In Table I, it is mentioned that S1PR5 is located in Intracellular membrane. Does this means intracellular membranes, like some organelles, or the inner left of the plasma membrane?

Lines 81 to 97 mention some information about the distinct S1P receptors, but this paragraph is difficult to understand, since it is quite fragmented. Authors should consider the possibility to add new columns on Table I to include the information of lines 81 to 97, and only include a very short presentation of the table in the text.

Line 92 (and others in the text) mentions the chemokine MCP1. Authors should mention also the name using recent nomenclature (CCL2).

Line 140 states that “Levels of sphingosine-1-phosphate (S1P) is increased…” and should say “are increased”.

Please include calcium channels (Receptor or Store-operated calcium channels) and the production of reactive oxygen species (ROS) in Figure 2.

Please include the process of FceRI crosslinking on Figure 2 and in the text. Receptor aggregation by the recognition of an antigen by at least two IgEs bound to the alpha subunit of the receptor is an essential step on the signaling cascade.

Please include the PKC-dependent activation of Bcl10 and MALT1 for NFkappaB activation on Figure2.

Figure 3 shows two molecules of IgE bound to a single alpha subunit of the FceRI receptor. Please include the dimerization of the receptor as an important step of its activation. One IgE should bind only one alpha subunit and allergen should be recognized by at least two IgE molecules.

Line 177 states that “Calcium ions can induce degranulation by increasing secretion of histamine….”  It should say “Calcium ions can induce degranulation, increasing the secretion of histamine…”?

Legend to figure 2 appears as part of the text?

Author Response

Dear Sir

Thanks for excellent suggestions. I made changes according to the suggestions.
In this revision, I sought help for English problems. I also send English certificate.   

Q/ This work provides general information about the sphingosine-1-phosphate and its receptors and its relationship with the signaling system of the high affinity IgE receptor in mast cells.   The text gives a general perspective of SP1 signaling pathways and suggests some cross-talk points with the FceRI receptor. The provided information is relevant since bioactive lipids have emerged as an important alternative to control inflammation. It is important to point out that the article exposes relevant information in the field of miRNAs and HDAC in the S1P system, which is quite novel and useful to the understanding of the long-term and complex actions of this lipid. However, information is not presented in a clear form, some concepts are repeated in distinct paragraphs and it is not possible to fully understand the message of the authors, since there is no discussion of the data. Some sections are composed of phrases exposing results of distinct papers and no final statement making an analysis or conclusion can be found. Finally, figures regarding the FceRI signal transduction system are quite incomplete and a number of elements should be included.

Ans. 1) I change figures 2,3, and 4 to accommodate your suggestions. Please take look at new figures. 2) In this revision, I try to remove unnecessary and redundant sentences. 3) In this revision, I try to rearrange sentences to make this manuscript more readable. 4) In this revision, I try to be more specific by giving discussion on data. 5) I try to fix English problem. I sought help from English editing service. I send English certificate. 

Specific examples:

Q/ In several parts of the review, main findings are mentioned without a clear context or conclusions, and some details on MC activation are duplicated. For example, section about Anaphylaxis and Sphingolipids starts with a very brief description of the signal transduction system of FceRI receptor but this description is not connected with the induction of anaphylaxis. Description of anaphylaxis appears until line 250 and lines 251 to 254 mention the role of pro-inflammatory mediators on the induction of allergic asthma, atopic dermatitis (which was already mentioned on a previous section).

Ans. I add new sentences to describe connection between FcεRI signaling and anaphylaxis: FcεRI is necessary for passive cutaneous anaphylaxis (PCA) reaction by activating MAPK signaling [83]. Allergen binds to at least two molecules of IgE (Figure 2). IgE then binds to alpha subunit of FcεRI and results in cross linking of FcεRI (Figure 2). Please take look at new manuscript (lines 181-183).

I add new sentence (lines 225-226): MCs can mediate various allergic reaction by secreting pro-inflammatory mediators [99, 100]. I agree some details on MC activation are duplicated. Figure 2 describes FcεRI signaling in general. Figure 3 describes cross talk between FcεRI and S1P signaling. To describe FcεRI signaling, there is some overlap. In this revision, I change figure 3 legend to avoid overlap (lines 213-220).    

Q/Lines 205 to 210 mention few references in which SIP/SPHK1 axis is explored but there is no conclusion of the presented information, since line 211 only states that “There is a role of S1P in anaphylaxis”

Ans. Thanks. I agree. I made changes to make it more readable. Please take look at these sentences: IgE-treated MCs showed increased levels of S1P and sphingosine-1-phopsphate receptor 3 (S1PR3) [89]. S1P is also necessary for calcium influx to activate PKC for MC granulation in the mouse model of PSA [90]. Mice lacking both the isoforms of SPHKs with undetectable level of circulating S1P show impaired survival after anaphylactic reaction [91]. I describe the role of S1P (90). Please take look at new manuscript (lines 190-193). 

Q/Line 213 mentions some data on the importance of COX2 expression, but data were obtained in cardiac fibroblasts and colon carcinogenesis. Line 217 contains information on the potential metastatic potential of breast cancer cells, but it is not clear the relationship of those data with S1P effects on Mast Cells. 

Ans. I agree. I made changes to make sentences more readable. I mention role of COX-2 in PSA. I mention that S1P increases COX-2 expression. also mention that allergic inflammation such as PSA can enhance metastatic potential of cancer cells. Based on these reports, it is probable that S1P and COX-2 forms positive feedback loop to mediate allergic inflammations. Please take look at new manuscript (Lines 195-205).             

Q/Data on metformin effects on passive cutaneous anaphylaxis (PCA) mentioned on lines 221-223 are not discussed, since authors only mentions that the cross-talk S1P and FceRI plays a critical role on anaphylaxis.

Ans. I agree. I add this sentence: Metformin inhibits FcεRI-mediated degranulation, IL-13, and S1P secretion in bone marrow-derived mast cells (BMMCs) [97]. Please take look at new manuscript (lines 206-207).

Q/Main ideas on section 6 are not easily to follow. It starts with reactions mediated by MCs (information that was already mentioned in previous sections), then DNA deacetylases are mentioned but the specific function of those enzymes on allergic reactions is not mentioned or discussed. In line 265, some information about lung tumors is presented but there is no explanation about the relationship between lung tumors, DNA deacetylases and allergies.

Ans. I agree. I think DNA deacetylases are histone deacetylases. As far as lung tumors are concerned, I add new sentence: miR-384 targets HDAC3 and suppresses positive feedback relationship between anaphylaxis and anaphylaxis-enhanced metastatic potential of cancer cells (line 237-239). I remove sentence that contains lung tumors. 

In this revision, I mention the role of HDAC6 in allergic inflammations (lines 251-256). I also describe role of HDAC3 in allergic inflammations (229-239).             

Q/Ideas exposed in lines 266 to 271 are not very clear. The experimental model and conditions in which studies were made are quite important here, since no relationship with allergies can be appreciated.

Ans. I thank for excellent suggestions. I changed sentences to make them more readable. I mention that TGaseII, increased by antigen stimulation during allergic inflammation, is responsible for the increased expression of HDAC3, PGE2 synthase, and increased level of ROS. This indicates probable role of TGase II in allergic inflammation. There has not been report on the role of TGase II in S1P production, However, it is probable that TGase II mediates allergic inflammations by increasing S1P production. I also mention that HDAC6 can increase S1P production based on previous reports that indicate role of HDAC6 in allergic inflammations. Please take look at new manuscript (lines 246-256).     

Q/Section 7 contains information already included in previous paragraphs and it is not clear what is the message and, more importantly, what is the conclusion? Detailed analysis and discussion must be included in this section.

Ans. I agree. In this revision, I try to make section 7 more readable. I add new sentence The SPHK1/2 or SPHK2 inhibitor can promote the activity of HDAC1 and inhibit the histone acetylation of the Krüppel-like factor 4 (KLF4) promoter regions to regulate M1 to M2 microglial polarization [113].

I also add this sentence [These reports suggest a relationship between S1P level and HDAC activity].

I want to stress that cross talk between FcεRI and S1P signaling mediates allergic inflammations through its effect on HDACs. In section 7, I also want to stress that HDACs are necessary for the increased expression of S1P during allergic inflammations, suggesting cross talk between S1P signaling and HDACs. Please take look sentence below.

The use of selective inhibitors of HDAC6 is under investigation as a potential treatment strategy for inflammatory diseases owing to their ability to regulate SPT, inflammatory cells and cytokines [118].   

Specific comments:

Q/Lines 50 and 51 should be part of Figure 1 Legend.

Ans. It was my mistake. I made changes as you suggested. Please take look at new manuscript (lines 44-45).    

Q/FceR1 must be written with a the latin number one (I)

Ans. Thanks I made changes as you suggested. Please take look at new table 1. 

Q/In Table I, it is mentioned that S1PR5 is located in Intracellular membrane. Does this means intracellular membranes, like some organelles, or the inner left of the plasma membrane?

Ans. I agree. I made mistakes. I remove intracellular membrane. Please take look at new table 1.   

Q/Lines 81 to 97 mention some information about the distinct S1P receptors, but this paragraph is difficult to understand, since it is quite fragmented. Authors should consider the possibility to add new columns on Table I to include the information of lines 81 to 97, and only include a very short presentation of the table in the text.

Ans. Thanks. I agree. I made changes as you suggested. Please take look at new manuscript (lines 70-86). In this revision, I add information on cells that express S1PRs. Please take look at new table 1. I add new sentences to mention role of S1PR2 (line 74). I try to paragraph to make this manuscript more readable.              

Q/Line 92 (and others in the text) mentions the chemokine MCP1. Authors should mention also the name using recent nomenclature (CCL2).

Ans. Thanks. I made change as you suggested. Please take look at new manuscript (line 230).       

Q/Line 140 states that “Levels of sphingosine-1-phosphate (S1P) is increased…” and should say “are increased”.

Ans. Thanks. I change it as you suggested. I change it into The S1P levels were observed to be higher in the BAL fluids of patients with asthma [68]. Please take look at new manuscript (lines 132-133).     

Q/Please include calcium channels (Receptor or Store-operated calcium channels) and the production of reactive oxygen species (ROS) in Figure 2.

Ans. New figure 2 shows calcium channels and the binding of IP3- to- IP3 receptor on ER. I indicate targets of ROS. Please take look at new figure 2. In my opinion, it would be difficult to know at which step ROS production takes place.            

Q/Please include the process of FceRI crosslinking on Figure 2 and in the text. Receptor aggregation by the recognition of an antigen by at least two IgEs bound to the alpha subunit of the receptor is an essential step on the signaling cascade.

Ans. Thanks. I agree. I change figure 2 as you suggested. I add this sentence [Allergen can be bound by two molecules of IgE, resulting in dimerization of FcεRI]. Please take look at new manuscript (lines 213-214).

Q/Please include the PKC-dependent activation of Bcl10 and MALT1 for NFkappaB activation on Figure2.

Ans. Thanks. I agree. I add BCL10, MALT1, and NF-kB in figure 2. Please take look at new figure 2.

Q/Figure 3 shows two molecules of IgE bound to a single alpha subunit of the FceRI receptor. Please include the dimerization of the receptor as an important step of its activation. One IgE should bind only one alpha subunit and allergen should be recognized by at least two IgE molecules.

Ans. Thanks for good suggestion. I change the figure as you suggested. I add this sentence [Allergen can be bound by two molecules of IgE which results in dimerization of FcεRI]. Please take look at new manuscript (lines 213-214).

Q/Line 177 states that “Calcium ions can induce degranulation by increasing secretion of histamine….”  It should say “Calcium ions can induce degranulation, increasing the secretion of histamine…”?

Ans. Thanks. I change as you suggested. Please take look at new manuscript (lines 162-163).   

Q/Legend to figure 2 appears as part of the text?

Ans. Thanks. If possible, I would like to legend to figure 2 as it is. I hope that this does not cause trouble. 

Round 2

Reviewer 1 Report

The authors have responded appropriately to the issues I have raised. There is nothing further to point out about this article.

Author Response

The authors have responded appropriately to the issues I have raised. There is nothing further to point out about this article.

Dear Sir

I thank for your generousness.

Sincerely yours

Reviewer 2 Report

Changes made by the authors have importantly improved the review. 

Section regarding the relationship between cancer, S1P and the FceRI receptor is not very clear, since the relationship between inflammation (as a general immune response) and cancer is not clearly explained, and the mentioned references are not specifically related to allergy and cancer. 

Several sentences presenting data from distinct studies are placed in the same paragraph but no conclusion is made of those observations.

Minor comments:

Please define PSA and PCA the first time terms are used.

Please define COX1 the first time is used and use COX1 consistently along the text.

Author Response

Changes made by the authors have importantly improved the review. 

Dear Sir

Thanks for excellent suggestions. I made changes according to the suggestions made by you. I hope that changes I made are fine. 

Sincerely yours

Q. Section regarding the relationship between cancer, S1P and the FceRI receptor is not very clear, since the relationship between inflammation (as a general immune response) and cancer is not clearly explained, and the mentioned references are not specifically related to allergy and cancer. 

Ans. Thanks. I agree. In this revision, I made changes to section 6 (FcεRI/S1P/HDACs in Anaphylaxis).

- In this revision, I try to make each paragraph of section 6 more readable.

- I first try to mention relationship between HDAC3 and MCP1. I add this sentence: HDAC3 directly increases MCP1 expression in antigen-stimulated RBL2H3 cells [101]. Please take a look line 245.

- I delete unnecessary sentences to make each paragraph of section 6 more readable. Please take a look new manuscript.

- I delete unrelated references and add new references (108, 109). These new references describe role of M2 macrophages polarization in mediating S1P signaling. We and others previously reported that M2 macrophages polarization induced by mast cells was necessary for enhanced tumorigenic and metastatic potential of cancer cells by PSA (Kwon Y et al., 2021; Kim M et al., 2021; Kwon Y et al., 2020; Eom SA et al., 2014). Throughout this revision, I remove unrelated references. For example, I add new reference 43. I delete ref. 136.         

- We and others previously reported that passive systemic anaphylaxis (PSA) enhances tumorigenic and metastatic potential of cancer cells by promoting cellular interactions between cancer cells, mast cells, and macrophages (Kwon Y et al., 2021; Kim M et al., 2021; Kwon Y et al., 2020; Eom S et al., 2014). We and others also showed that mast cells can activate cancer cells, macrophages, and endothelial cells. These reports suggest close relationship between anaphylaxis and cancer.   

* I add this sentence: We previously reported that PSA enhanced the tumorigenic and metastatic      potential of mouse melanoma cells through the induction of HDAC3, MCP1, and CD11b (a macrophage marker) expressions [102].

* I add this sentence: It is probable that extracellular MCP1 may promote these cellular interactions. Thus, HDAC3-MCP1 axis mediates anaphylaxis-enhanced tumorigenic and metastatic potential by promoting cellular interactions.

* I also add this sentence: HDAC6 is necessary for enhanced tumorigenic potential of melanoma cells by PSA [106]. HDAC6 promotes cellular interactions by increasing IL-27 during anaphylaxis [106]. Thus, HDAC6-IL-27 axis may increase S1P production during allergic inflammations.

* I hope that the above sentences stress relationship between allergy and cancer.  

Q. Several sentences presenting data from distinct studies are placed in the same paragraph but no conclusion is made of those observations.

Ans. Thanks. I agree. In this revision, I try to make conclusion in each paragraph of section 6. Please take a look section 6 of new manuscript. In this revision, I change each paragraph to contain sentences that are connected in a way to convey purpose of each paragraph. For that, I add new sentences and remove unnecessary sentences. I also make conclusion at the end of each paragraph in every section. Please take a look new manuscript.        

Minor comments:

Q. Please define PSA and PCA the first time terms are used.

Ans. Thanks. I define PSA the first time the terms are used; please take a look at line 204.

I define PCA the first time the terms are used; please take a look line 192.

Q. Please define COX1 the first time is used and use COX1 consistently along the text.

Ans. Thanks. Do you mean COX2? I did not mention COX1 in this manuscript. I define COX2; please take a look lines 71-72.